# The Garden of Forking Paths:* Observing Dynamic Parameters Distribution in Large Language Models

## Abstract

A substantial gap persists in understanding the reasons behind the exceptional performance of the Transformer architecture in NLP. A particularly unexplored area involves the mechanistic description of how the distribution of parameters evolves over time during training. In this work we suggest that looking at the time evolution of the statistic distribution of model parameters, and specifically at *bifurcation* effects, can help understanding the model quality, potentially reducing training costs and evaluation efforts and empirically showing the reasons behind the effectiveness of weights sparsification.

## 1 Introduction

Since its introduction in 2017 (Vaswani et al., 2017), the Transformer architecture has spurred enormous research efforts leading to significant advancements across many research fields such as Computer Vision (Dosovitskiy et al., 2020), Speech (Gulati et al., 2020) and Language Processing (Brown et al., 2020). As the deployment of state-of-the-art language models becomes ubiquitous in natural language processing applications, the demand for transparency and explainability has intensified and a new research area denoted as Mechanistic Interpretability (MI) emerged (Conmy et al., 2023).

MI is the attempt to microscopically describe the internals of neural networks by analysing the weights, with the goal of reverse engineering their macroscopic properties. Similar to how statistical mechanics links microscopic particle behavior to macroscopic system properties, MI delves into the micro-level details of neural networks' parameters to elucidate their impact on macro-level functionalities and model outputs. This analogy underscores interpretability's role in bridging the explanatory gap within neural networks, akin to how statistical mechanics contributes to understanding collective dynamics in particle systems where individual microscopic laws give rise to large scale properties (Huang, 2008).

In this sense, researchers have undertaken diverse approaches, ranging from probing attention mechanisms (Gurnee & Tegmark, 2023) to understand the internal space and time representation of Large Language Models (LLMs), or by analyzing residual streams (Yu & Yang, 2023) as a way to describe the concepts flowing through the network's layers. Others have pushed forward analogies between Transformers and interacting particle systems (Geshkovski et al., 2023), where each word, akin to a particle in a ensemble, follows the flow influenced by the collective behavior.

In this paper, we extend the analogy between Mechanistic Interpretability and Statistical Mechanics (Huang, 2008) to investigate the microscopic network parameters' evolution over time, by exploring the properties of *Pythia* (Biderman et al., 2023), a publicly accessible LLM. We describe the model in terms of dynamical parameters' evolution of the embedding layers, and study their effect on the generated output.

Our empirical findings suggest a two-fold character of these internal parameters' dynamics:

- In the first phases of the training process, a diffusive process takes place where the model explores the landscape in every direction (as many as the number of network parameters);

---

*"The Garden of Forking Paths" is a tribute to the title of a short novel from the writer and poet Jorge Luis Borges.

- Then, after a certain transient period, the parameters' dynamic converges to a deterministic evolution, whereas the underlying process is no longer diffusive.

With this study we aim to shed light on unexpected dynamics unfolding beyond the transient period. We posit that this phenomenon is profound and has far-reaching implications, thus demanding thorough analyses. Moreover, the observed phenomenon of weights converging to two distinct, apparently zero-symmetric values, warrants further investigation due to its potential biological relevance. This symmetry might hint at an underlying process mirrored in the brain, where excitatory and inhibitory neurons compete for dominance during learning. Indeed, as observed in Najafi et al. (2020), excitatory and inhibitory subnetworks are equally selective during decision-making process, and emerge simultaneously during learning process. This intriguing parallelism demands deeper analysis to illuminate the connection between our forking path behavior, observed in diverse model sizes, and the corresponding neurological processes observed in the human and animal brain (Najafi et al., 2020).

Our observations yield insights with significant practical applications, particularly in relation to the dynamics observed during the training processes of neural networks. The presence of a bifurcation phenomenon within the dynamics of the weights—across different models of varying sizes and trained on diverse datasets—naturally suggests a practical protocol for spontaneously stopping the training process. Specifically, this bifurcation signals a transition to a stationary state, indicating that further training may not alter the weight values significantly. Therefore, by recognizing the absence of significant fluctuations in the dynamics, one can efficiently conclude the training once such a stationary state is achieved. This advancement has profound implications for efforts to mitigate the environmental impact of training LLMs. In an era where combating climate change is paramount, reducing the energy consumption of training processes is a critical goal. Our new exit protocol for training offers a strategy to achieve this objective, minimizing energy expenditure without compromising the effectiveness of the training.

The structure of this manuscript follows a logical progression to facilitate the understanding of our work. In Section 3 we provide an overview of the internal architecture of LLMs. We analyze and detail the methods and tools we utilized to investigate the dynamics of the last embedding layer parameters. Following this, in Section 4 we outline our key findings, with the effects of the bifurcation on model perplexity. We provide a possible interpretation of these phenomena, aligning them with the underlying theory, and suggest possible causes and implications. We finally summarize our insights, reiterate the implications of our findings, and indicate directions for future research.

## 2 Related work

Our work follows the direction traced by Mechanistic Interpretability (MI) studies recently done for the Transformers architecture. The first supporting studies behind most MI attempts are based on intuitive visualizations of the internal layers of Transformers (Vig, 2019; Voita et al., 2019; Chefer et al., 2021), with most of them based on individual neurons as unit of analysis.

At the heart of MI lies the conjecture that artificial neural networks, akin to their biological counterparts, exhibit a nuanced interdependence between structure and function (Sporns, 2016). The algorithms implicitly encapsulated within the computational graphs of models (analogous to synapses in the brain) are intricately linked to synaptic strengths (parameter values). Consequently, the foundational capabilities manifested by both systems are contingent upon the synergy of these inherent structural and functional attributes (Liu et al., 2024; Nainani, 2024). In most MI studies it is prevailing to identify the neurons of a model as the fundamental unit of examination. However, a scrutiny centered on neurons may lack insightfulness due to phenomena such as *polysemanticity*, i.e. the neurons' capacity to exhibit distinct responses to unrelated inputs, as elucidated in previous works (Olah et al., 2017; Bricken et al., 2023).

Our work tries to address some of the aspects exposed in the MI literature. In particular, to our knowledge, we are the first to visually analyze the parameters distribution of the embedding layers among multiple training checkpoints, individuating a bifurcation effect with a quasi-symmetric bimodal weights distribution. We believe our observation is at the root of the effectiveness of extreme quantization methods like

the one recently proposed by Ma et al. (2024) where weights are allowed to only take 3 distinct values $\{1, 0, -1\}$.

Powerful MI techniques are used to explain factual association and hallucinations in Meng et al. (2022) and Yuksekgonul et al. (2023), where the authors focused at the individual parameter-level showing that while each MLP layer in the Transformer block has the ability to store factual associations, the attention layer acts more like a router, transferring factual knowledge where it will be used in the next layer. Moreover, as shown by Voita et al. (2023), having a full mechanistic interpretation of the evolution of network parameters is important, as fully trained models display a large amount of neurons that never activate (dead neurons).

These observations motivate our study, as one of the reasons why methods like quantization (Dettmers & Zettlemoyer, 2023; Xiao et al., 2023) and sparsification (Dettmers et al., 2022) lead to good downstream results, is *probably* a decrease in network information during training. Specifically, as demonstrated by Achille et al. (2017), in the initial part of training a network weights are highly sensitive to input data and tend to contain less information. Once the connections are aligned with the data distribution, they are harder to modify, an observation that is also linked to learning processes in animals and human (Najafi et al., 2020). In this respect, our work accumulates further experimental evidence about the presence of dead neurons caused by counteracting inhibitory and excitatory effects.

Furthermore, the curious phenomenon of grokking (Power et al., 2022) is analyzed with the lens of MI in Nanda et al. (2023) where the sudden increase in validation accuracy is explained as a gradual (and not sudden) amplification of individual neural mechanisms encoded in the network weights. Other works such as the ones by Liu et al. (2022a;b) have instead leveraged statistical mechanics principles to decode the intricate *grokking* behavior witnessed in deep learning models. The observations of our study are in line with the questions raised in the work by Merrill et al. (2023): how does grokking relate to network sparsification? In other words, are training set memorization (as in grokking) and extreme network self-sparsification two aspects of the same phenomenon? Indeed, grokking can be seen as the competition of a dense network that dominates before the transition and generalizes poorly, and a sparse one that dominates afterwards (Merrill et al., 2023).

The emerging properties of LLMs – and the lack thereof (Wei et al., 2022) – can also be evaluated in MI terms. For example, the work by Schaeffer et al. (2023) has shown that emergent abilities are heavily dependent on the non-linearity of the researcher's choice of evaluation metrics, rather than specific phenomena within network weights. Indeed, when evaluating the emergence of abilities using discrete evaluation metrics (on multiple answers datasets) any improvement is detected only when exceeding a random choice threshold, thus giving the illusion of 'emergence' while instead the answer is simply efficiently retrieved from the pre-training weights (Lu et al., 2023). This is the main reason why we only concentrate on perplexity measurements (Section 4.2) rather than other evaluation metrics.

Motivated by these theoretical and empirical investigations in the next sections we detail our methodological approach and results.

## 3 Materials and methods

### 3.1 Models and data

We analyzed the 143 checkpoints of the well-known LLM Pythia (Biderman et al., 2023), trained on both the deduplicated and non-deduplicated *ThePile* dataset (Gao et al., 2020) and made available by EleutherAI[1] through the Huggingface platform (Wolf et al., 2019) to facilitate interpretability works in both spatial and temporal scaling dimensions. In pursuit of reproducibility, the Pythia model suite ensures uniformity across its networks by employing identical global architectures (see Section 3.2), utilizing the same optimization method, and processing data from a consistent dataset in a standardized order.

---

[1]https://www.eleuther.ai

The network size ranges from small ($14M$ parameters) to very large ($\approx 12B$ parameters), with the number of layers ranging from a minimum of six for the 14M, 31M and 70M models, to a maximum of thirty-six for the largest 12B model.

Importantly, we have utilized two sets of models and we indicate in the text whether the model was trained on the deduplicated (DD) or non-deduplicated (NDD) *ThePile* dataset. More specifically, the smallest models from the Pythia suite ($14M$ and $31M$ parameters) have been trained on the NDD dataset, while the models from 70M parameters and above are instead trained on the DD dataset. We analyzed both the NDD and DD models: while the underlying training dataset is different, the behaviour observed on their dynamics is similar.

As a side note, although intermediate checkpoints have been disclosed for BLOOM as well (Workshop et al., 2022), BLOOM has undergone training using a singular model size, specifically with 176 billion parameters. In contrast, Pythia has been trained across a spectrum of model sizes. This distinctive aspect, facilitating examinations of both architectural scaling and training dynamics, serves as the rationale behind our preference for Pythia over BLOOM. Finally, due to the high computational costs, this work focuses on the smallest models, with parameters count ranging from 14M to 1B, as indicated in Table 1 by boldface rows.

## 3.2 Network architecture

We follow the notation delineated in McGrath et al. (2023). Transformers function by processing sequences of tensors flowing through a series of self-attention operations and token-wise feed-forward layers. Mathematically, an auto-regressive language model is a map from $t-1$ input tokens $x_{<t} = (x_1, \ldots, x_{t-1})$ to a probability distribution over the next token $x_t$ using a function $f_\theta$:

$$p(x_t|x_{<t}) = f_{\boldsymbol{\theta}}(x_{<t}) \tag{1}$$
$$= \text{softmax}\left(\boldsymbol{\pi}_t(x_{<t})\right) \tag{2}$$

where $\theta$ are the network parameters, distributed in multiple blocks and layers, and the output token scores $\boldsymbol{\pi}_t$ are called *logits*. The network architecture is encoded by the recursive function $f_\theta$, composed of a first embedding layer mapping tokens into the latent network space (with an additional matrix of learnable positional embeddings $\mathbf{W}_P$), and followed by $L$ repeated stacked layers implementing the following recursive operations:

$$\boldsymbol{\pi}_t = \text{LayerNorm}(\mathbf{z}_t^l) \cdot \mathbf{W}_U$$
$$\mathbf{z}_t^l = z_{t-1}^l + \mathbf{a}_t^l + \mathbf{m}_t^l$$
$$\mathbf{a}_t^l = \text{MHSA}(\mathbf{z}_{\leq t}^{l-1})$$
$$\mathbf{m}_t^l = \text{MLP}(\mathbf{z}_t^{l-1}) \tag{3}$$

where `LayerNorm` is the layer normalization operation (Ba et al., 2016), MHSA is the Multi Head Self Attention operator (Vaswani et al., 2017) and $\text{MLP}(\cdot)$ is a two layer perceptron with GeLU activation function.

Our emphasis is on decoder-only, autoregressive language models employing a causal attention mask. Specifically, the Pythia models (Biderman et al., 2023) are based on the GPT-NeoX architecture with a few modifications such as the introduction of rotary embeddings for the matrix $\mathbf{W}_P$ (Su et al., 2021) and untied [2] embedding and unembedding matrices (Belrose et al., 2023), coloured in green in Figure 1. Additionally, the basic Transformer block in Pythia entails a MHSA operator implemented through *FlashAttention* (Dao et al., 2022) for computational efficiency reasons. To be precise, while the inner details of the GPT-NeoX architecture are slightly different from the description provided in the recursive set of rules indicated in Eq. 3, the main logic remains largely the same.

---

[2]As opposed to tied embedding and unembedding matrices where $W_U^T = W_E$.

The last unembedding layer is represented by a rectangular matrix $\mathbf{W}_U$ mapping the latent embedding space into the larger vocabulary space. Finally, the softmax operation converts the *logits* $\boldsymbol{\pi}_t$ to properly normalized probabilities. The properties of the analyzed models in the Pythia suite utilized in this paper are reported in Table 1.

| Model Size | Non-embed. parameters | Layers | Embed dimension | Heads | Dataset |
|------------|-----------------------|--------|-----------------|-------|---------|
| **14 M**   | $1.2M$  | 6  | 128  | 8  | NDD |
| **31 M**   | $4.7M$  | 6  | 256  | 8  | NDD |
| **70 M**   | $19M$   | 6  | 512  | 8  | DD  |
| **160 M**  | $85M$   | 12 | 768  | 12 | DD  |
| **410 M**  | $302M$  | 24 | 1024 | 16 | DD  |
| **1.0B**   | $806M$  | 16 | 2048 | 8  | DD  |

Table 1: Properties of the Pythia models utilized in this work.

Embedding parameters' count is the product of embedding dimension times the vocabulary size and has a larger effect in smaller and shallower models, whereas in larger models this is proportionally less relevant. The entire Pythia decoder-only network architecture is shown in Figure 1.

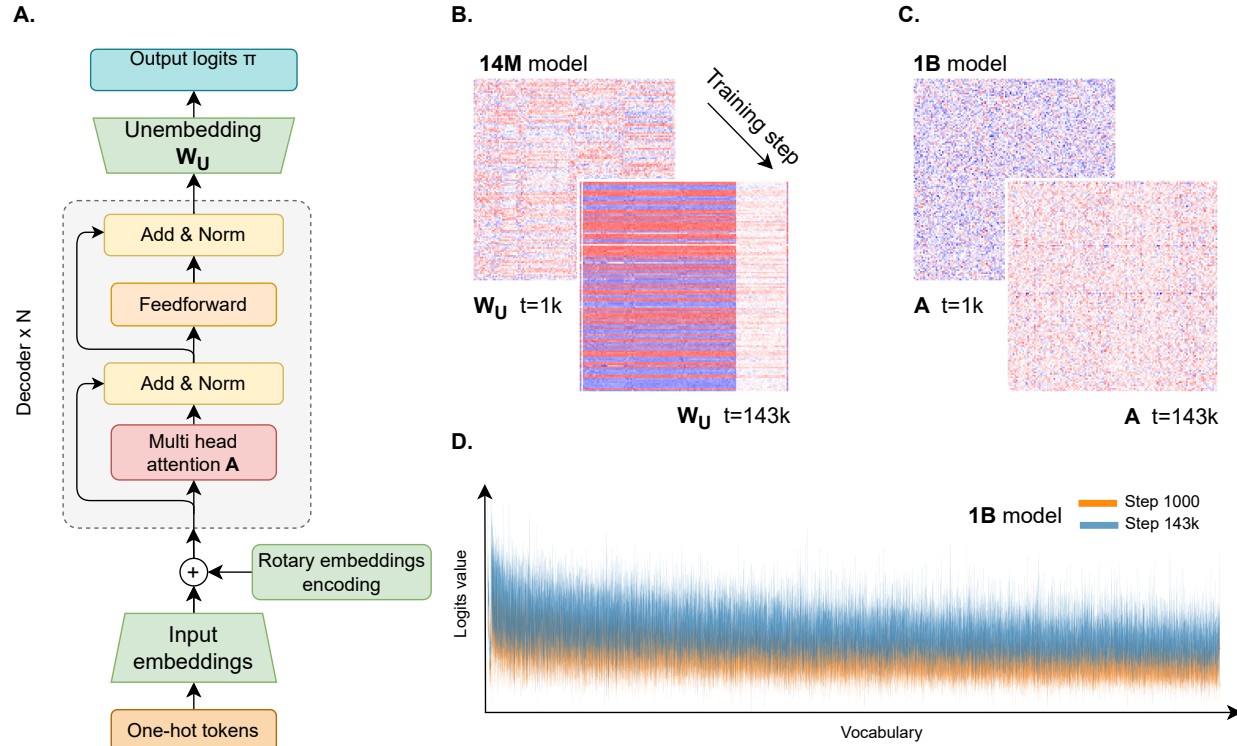

Figure 1: Panel **A.** Pythia models' basic architecture. The unembedding layer $\mathbf{W}_U$ is the last green layer. The attention matrix **A** is entailed in the red coloured multi-head self attention layer. Panel **B.** shows the output embedding matrix at first and last training step for the 14M model. For illustration purpose the first 512 out of 50304 columns are shown. Panel **C.** shows the first layer attention matrix at first and last training step for the 1B model. Panel **D** shows the average of token logits of a long sentence for the 1B model both at first and last training step.

### 3.2.1 Dealing with very large arrays

Each model includes 143 training checkpoints at constant intervals of 1000 steps – from step 1000 to 143,000. Storing in memory, for instance the 70M model, is feasible, requiring approximately 24GB of RAM for the full snapshot in `float16` precision; however, this proves unfeasible with larger models due to rapidly escalating memory requirements. For this reason, we have selectively sliced specific layers of the network, while avoiding explicit storage in memory operating through disk/memory mapping. This memory mapping operation allows to visualize the temporal evolution over the training dimension of specific subsets of the model parameters.

## 4 Results

In this section, we provide a number of interesting observations about the temporal dynamics of individual network parameters. We have focused our analysis on the final unembedding layer represented by $\mathbf{W}_U$, a large rectangular matrix mapping the latent embedding space into the vocabulary space (and viceversa for the initial embedding layer). The matrix $\mathbf{W}_U$ has shape $(d, v)$, where $d$ is the embedding dimension specific to each model, ranging from 128 to 2048, and $v = 50304$ is the fixed vocabulary size of the BPE tokenizer (Black et al., 2022). Each of the Pythia suite models, as described earlier, contains 143 checkpoints of $\mathbf{W}_U$. Importantly, the unembedding layer $\mathbf{W}_U$ provides a direct mapping from the latent embedding space to the more explainable dictionary space where each element is a token.

### 4.1 Temporal dynamic of unembedding layer parameters

We denote the entire history of unembedding layer matrices $\mathbf{W}_U$ with elements $w_{tdv}$, where: the first index $t$ denotes the checkpoint $1000 \leq t \leq 143000$; $d$ is the embedding dimension, and the $v$ is the vocabulary size. For visualization purposes, we flatten $w_{tdv}$ over the last two dimensions to obtain a matrix with two indices $t$ and $k$ where $k = 1, \ldots (d \times v)$.

Figure 2 shows the temporal evolution of the parameter density for the unembedding layer denoted as $w_{tk}$ for the 14M, 31M, 70M and 160M models, top left, top right, bottom left and bottom right panel respectively. 14M and 31M models were trained on the non-deduplicated (NDD) version of *ThePile* dataset, while 70M and 160M models were trained on the deduplicated (DD) version of *ThePile* dataset. Figure 2 shows how such models exhibit abrupt changes in their temporal dynamics. All models display two clear regimes: the first one is diffusive, the second one is bimodal quasi-deterministic. For example, in the bottom left panel, we observe how, for the 70M model, these two clear regimes emerge: before reaching $\approx 80,000$ training steps, the dynamics of the model's parameters resemble a diffusion process; conversely, after $\approx 80,000$ training steps, the weights move into a bimodal quasi-deterministic process. From the bottom right panel of Figure 2, we observe a very similar behavior for the 160M model, only shifted temporally along the training time axis, and thus emerging after $\approx 105,000$ training steps. The other two models, i.e., 14M and 31M, exhibit the same behavior; however, because they were trained on a different dataset, namely, the non-deduplicated (NDD) one, a comparison between the four models to understand a possible scaling with the number of training steps cannot be made and it will be addressed in future publications. Here, the only observation we can make is that the composition of the dataset might influence the timing of the bifurcation event (Ott, 2002). Specifically, when training involves non-deduplicated (NDD) datasets, the redundant information contained within may hinder the rapid approach to the bifurcation point.

The observation that the models' weights reach a stationary state through their own dynamics, as mentioned in the introduction, naturally suggests a practical protocol for spontaneously ending the training process: this bifurcation marks a transition to a stationary state, indicating that further training is unlikely to significantly alter the weight values. Thus, by observing the absence of significant fluctuations in the dynamics, one can efficiently terminate the training upon achieving such a stationary state. Moreover, our observations also suggest that a specific *quantization* of the weights can be performed. Indeed, as shown in Ma & et al. (2024), using a ternary set of $\{-1, 0, 1\}$ for each parameter, the performance of large language models (LLMs) remains unchanged. In our observations, particularly in the unembedding layer, a quantization to distinct values for the weights occurs naturally.

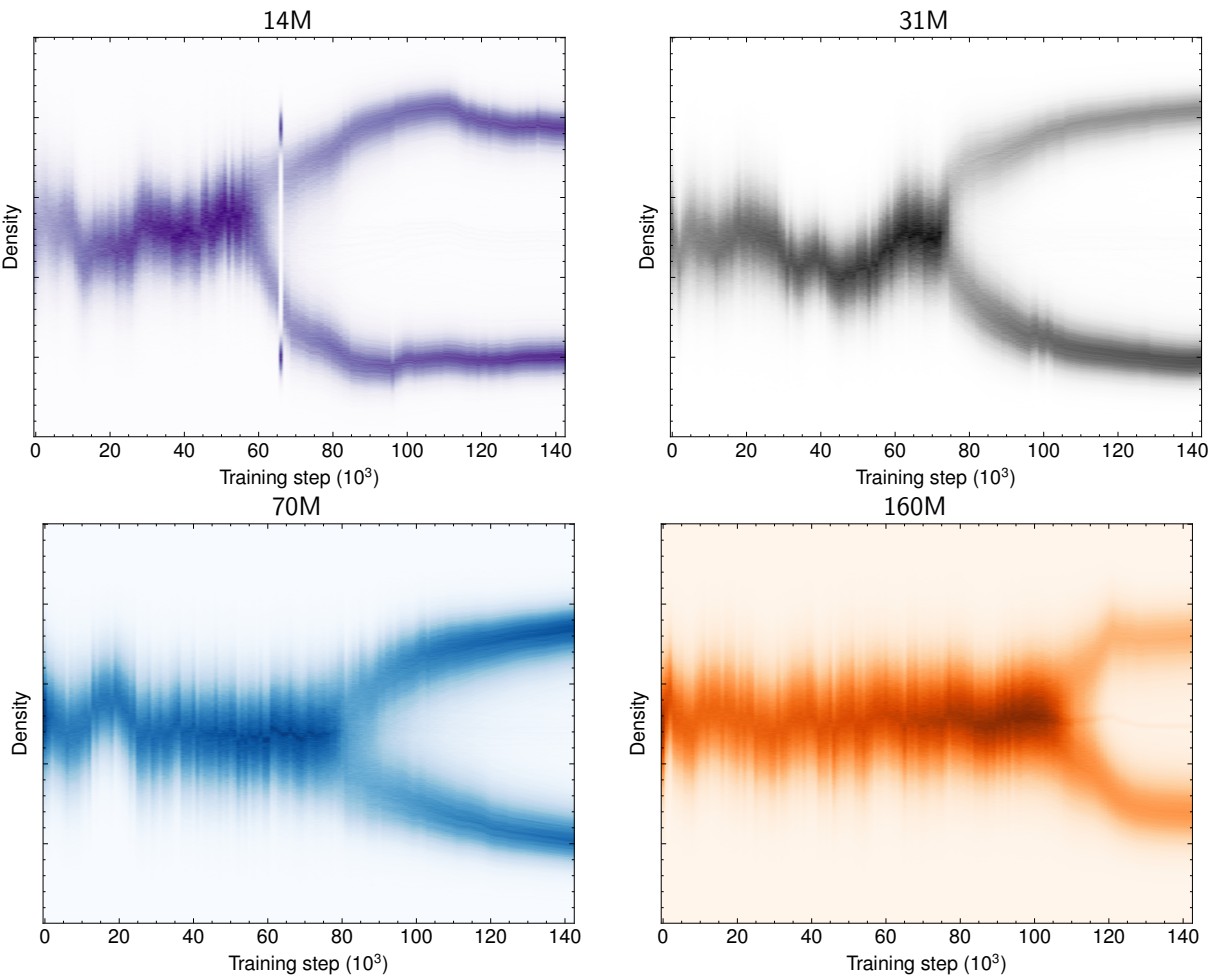

Figure 2: Dynamics of the density of the unembedding layer for four models. On the first row the models trained on NDD dataset, on the bottom row the models trained on DD dataset.

To better quantify the above observations, we have computed the mean square displacement over time for the models. To do this, we integrate the demeaned slices of $w_{tk}$ over the first dimension $t$ from 1 to $\tau$ obtaining a new array $\hat{w}$ as:

$$\hat{w}_{\tau k} = \sum_{t=1}^{\tau} w_{tk} - \langle w_{tk} \rangle, \tag{4}$$

where $\langle w_{tk} \rangle = \frac{1}{d \times v} \sum_{k=1}^{d \times v} w_{tk}$. We then indicate the variance of $\hat{w}_{\tau k}$ over each temporal slice as the mean square displacement $\mathrm{MSD}(\tau)$, defined as:

$$\mathrm{MSD}(\tau) = \frac{1}{(d \times v) - 1} \sum_{k=1}^{d \times v} \hat{w}_{\tau k}^2 \tag{5}$$

Figure 3 illustrates the evolution of $\mathrm{MSD}(\tau)$ over the available checkpoints for the 70M and 160M models as well for the 14M and 31M. In physics, Brownian motion (Uhlenbeck & Ornstein, 1930) displays a linear relation between mean-squared displacement and time (Zwanzig, 2001). In this case for both models, we observe a quasi-linear growth. We believe this has to do with unbounded brownian diffusion in a first phase of training, where uniform spreading of network parameters takes place within the loss landscape. However, surprisingly after a peak in $\mathrm{MSD}(\tau)$, corresponding exactly to the bifurcation (Strogatz, 2018)

discussed above, a sharp fall follows. The drastic decrease of MSD($\tau$) (to its own small value) is related to sub-linear diffusion, a phenomenon happening in the physics of crowded systems (Kok et al., 2016), where the network is approaching a narrow minimum in the loss function landscape.

We hypothesize that the above described bifurcation process will appear for the 410M model too, however we are restricted by the limited checkpoint availability. Hence, we reasonably believe that the bifurcation hypothesis is valid and list some observations supporting it in the next sections of this manuscript.

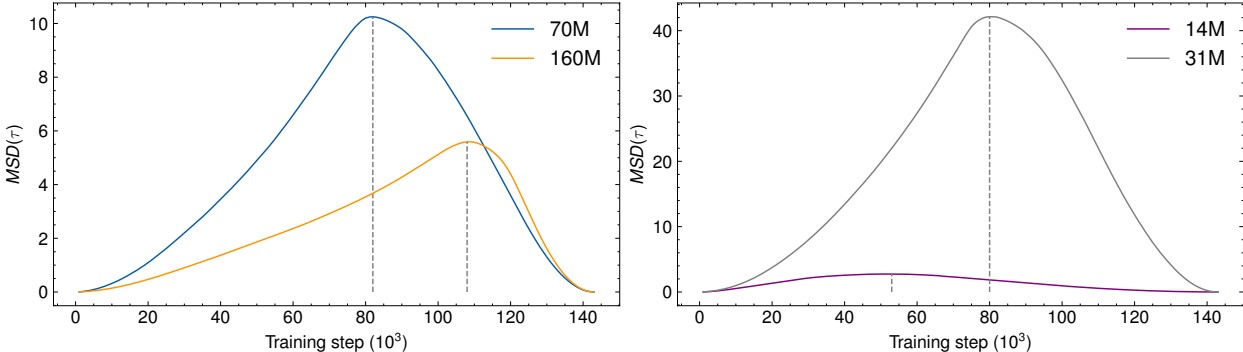

Figure 3: Mean square displacement over unembedding layer weights as a function of the training steps. **Left:** the smallest 70M and 160M on the deduped dataset. **Right:** the smallest models 14M and 31M on the non-deduped dataset. Vertical dashed lines are shown at the peak $MSD(\tau)$.

Importantly, the MSD($\tau$) is an implicit model property and it is not dependent on evaluation datasets. In the next sections we draw a parallel between the observed pattern of gradual increase and sudden decrease of MSD($\tau$) with model perplexity, suggesting that this numerical figure may significantly influence text generation tasks. This observation makes it possible to numerically predict the onset of better text quality without the need of standard evaluation metrics, like those based on multiple answer or those based on last word prediction (Wang et al., 2018; Paperno et al., 2016).

## 4.2 Evaluation of model perplexity

We have evaluated the model perplexity based on token logits of a series of fixed length phrases. Notwithstanding some known limitations, like direct effects of punctuation marks or dependency with text length (Wang et al., 2022), perplexity is widely considered a fair and intuitive metric and it is commonly adopted for language modeling evaluation. Indeed, as demonstrated in Dettmers & Zettlemoyer (2023), evaluations based on perplexity are sufficient and preferable for comparing text generation tasks.

Perplexity is defined for a sequence of $T$ tokens $\mathbf{X} = (x_0, x_1, \ldots, x_T)$ as the exponential of the average of negative log-likelihoods of a sequence of tokens. It reads:

$$PPL(\mathbf{X}) = \exp\left\{-\frac{1}{T}\sum_{i=1}^{T}\log p_\theta(x_t|x_{<i})\right\}, \tag{6}$$

where $p_\theta(x_t|x_{<t})$ is defined in Equation 1. Lower perplexity values indicate that the model is predicting the next token with high level of confidence, while high perplexity indicates that most of the tokens appear with equally likely probability, hence providing the text generation phase with high levels of ambiguity on the next token to predict.

### 4.2.1 Forward approach

We have evaluated the perplexity of entire sentences contained in the first 500 elements of the test set of the Lambada dataset (Paperno et al., 2016). We calculated the perplexity score for the above sentences across all the models mentioned in the text, considering different checkpoints and model sizes. The results,

shown in Figure 4, confirm the bifurcation behaviour observed over the unembedding layer both for the DD and NDD models. The effect is more evident when appreciated in the logarithmic scale (right panel, Figure 4). The model perplexity drops to zero exactly for the checkpoints where the bifurcation starts. We have limited the evaluation on 10% of the Lambada test set for computational reasons.

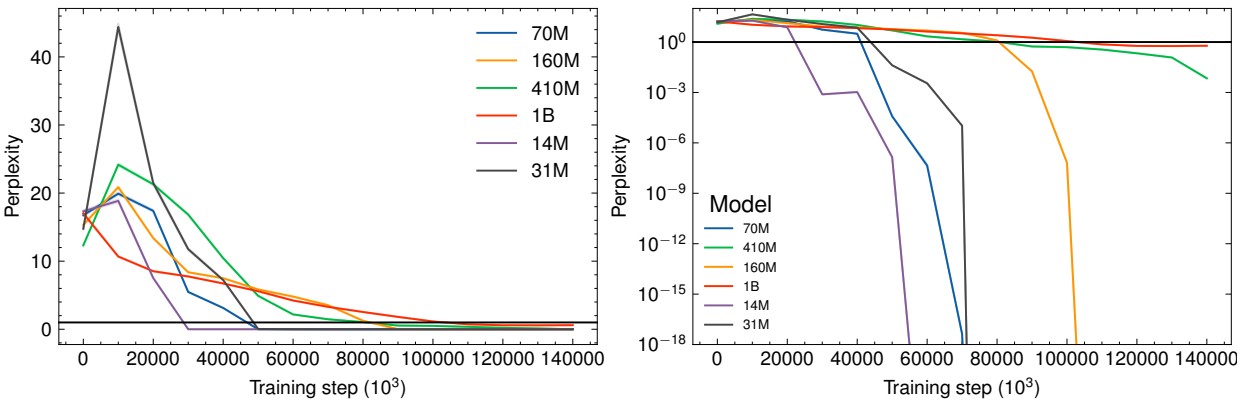

Figure 4: Perplexity of generated tokens on the first 500 examples of the test set of Lambada dataset. **Left:** perplexity expressed in linear scale. **Right:** same plot of perplexity but expressed in logarithmic scale. An horizontal black line is drawn at zero perplexity in both plots.

### 4.2.2 Causal unmasking approach

In parallel to the above simple perplexity calculation, we have devised another method to augment the dataset that takes into account the model generated text quality. In detail, we have run a text generation process with causal unmasking of the tokens in each phrase. We first tokenized each individual sentence **s** with the *GPTNeoxTokenizer*, obtaining a sequence of $t_s$ tokens $\mathbf{X}_s = (x_1, \ldots x_{t_s})$. For each tokenized sentence $\mathbf{X}_s$ we then considered the set of all $t_s$ linearly ordered sub-sentences ranging from length 1 to $t_s$, in other words the set $\mathbf{x}_{s_k} = \{(x_1, \ldots, x_k)\}$ with $k \leq t_s$. We then let the model complete each of the sub-sentences $\mathbf{x}_{s_k}$ up to the original length $t_s$, resulting in a total of $t_s$ sets of logits for each sentence $\mathbf{X}$.

At the initial phase with only few tokens being available, the model is forced to generate a large number of remaining tokens, with results that spawn from pure repetition of few tokens (in early phases of training) to repeated emission of longer sub-sequences. In this way we can evaluate how the model builds up on its internal knowledge, having only a few tokens at disposal. On the other hand, when approaching the end of the phrase, we expect logits to be more sharply peaked. For this reason we verified if logits are more sharply peaked around certain tokens. As expected, larger models tend to provide realistic texts even in the early phases of training, while smaller models have the tendency to repeat parts of the unmasked input.

We illustrate the causal unmasking process in Figure 5 with a short example phrase.

Similarly to what observed with the forward approach, Figure 6 shows a dramatic drop towards zero of the perplexity for both the 70M and 160M models exactly at the same checkpoints displaying the emergence of bifurcation in the parameters' plot and the drop of the diffusion coefficient as already shown in Figure 3. We also note that, as hypothesized above, perplexity for the 410M model is starting to drop to zero close to the last checkpoints. This behaviour supports our hypothesis that larger models should tend to exhibit the bifurcation behaviour, but only when trained longer than for the available checkpoints.

### 4.3 Rank of output embedding layer covariance

To support our findings, we have finally tested an hypothesis regarding the multidimensional distribution of the embedding vectors produced by the last output unembedding matrix for all models, but here we present the results only for the 70M and 160M models.

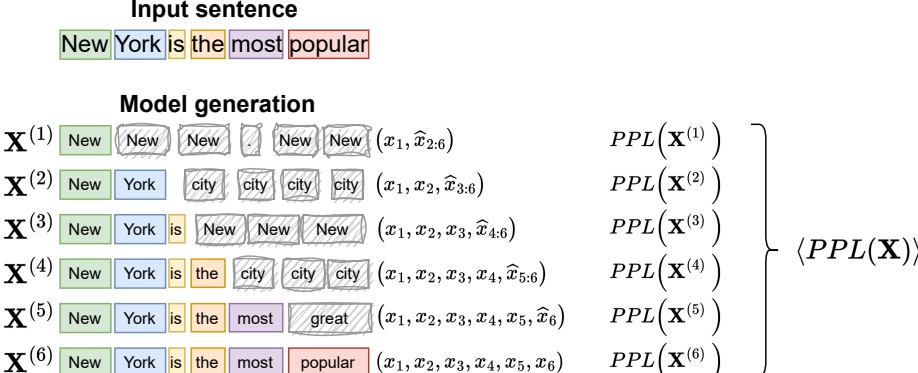

Figure 5: Causal unmasking process. The tokenized sentence is used to generate six new sentences, where the model completes from an initial set of tokens up to the initial phrase number of tokens. At each newly generated sub-sentence the model generates new tokens, depicted in gray.

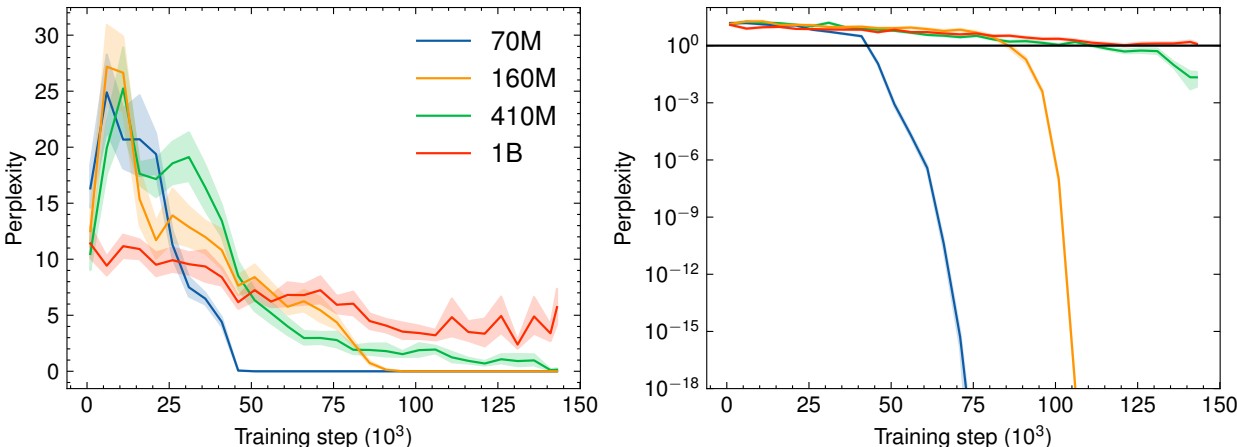

Figure 6: Perplexity of generated tokens with the causal unmasking approach. Left: perplexity expressed in linear scale. Right: same plot of perplexity but expressed in logarithmic scale. It is possible to see perplexity rapidly going to zero exactly at the point of maximum $MSD(\tau)$. The smaller models, 14M and 31M, are not shown in the figure solely for the sake of image clarity, but they show the same behaviour.

We have fed batches of various sizes (from 16 to 64) of multivariate normally distributed vectors $\mathbf{V} \sim \mathcal{N}(0, \mathbf{I}_{d_{\text{embed}}})$ and multiplied them with the output unembedding matrix $\mathbf{W}_{U_t}$ at different checkpoints to get transformed batches of vectors $\hat{\mathbf{V}}_t = \mathbf{W}_{U_t} \cdot \mathbf{V}$.

Then, we have computed the covariance matrices $\mathbf{C}_t$ over the batch dimension of the resulting vectors $\hat{\mathbf{V}}_t$ and evaluated rank$(\mathbf{C}_t)$ up to a tolerance parameter with values in the domain $[0.1, 1]$. The covariance matrix rank shrinks at different tolerance levels exactly at the checkpoints for which the model exhibits the bifurcation in the weights' dynamics as from Figure 2.

The net effect of the covariance matrix shrinkage observed in Figure 7 is the projection of isotropic embeddings onto a smaller subspace of the full vocabulary. This suggests that after the bifurcation, the model focuses on a subset of the possible emitted tokens, leaving a much smaller probability to tokens associated with dimensions for which the subspace is reduced, in accordance to the uneven distribution of words in natural language, which is known to follow a power-law distribution of words (Zipf, 1949).

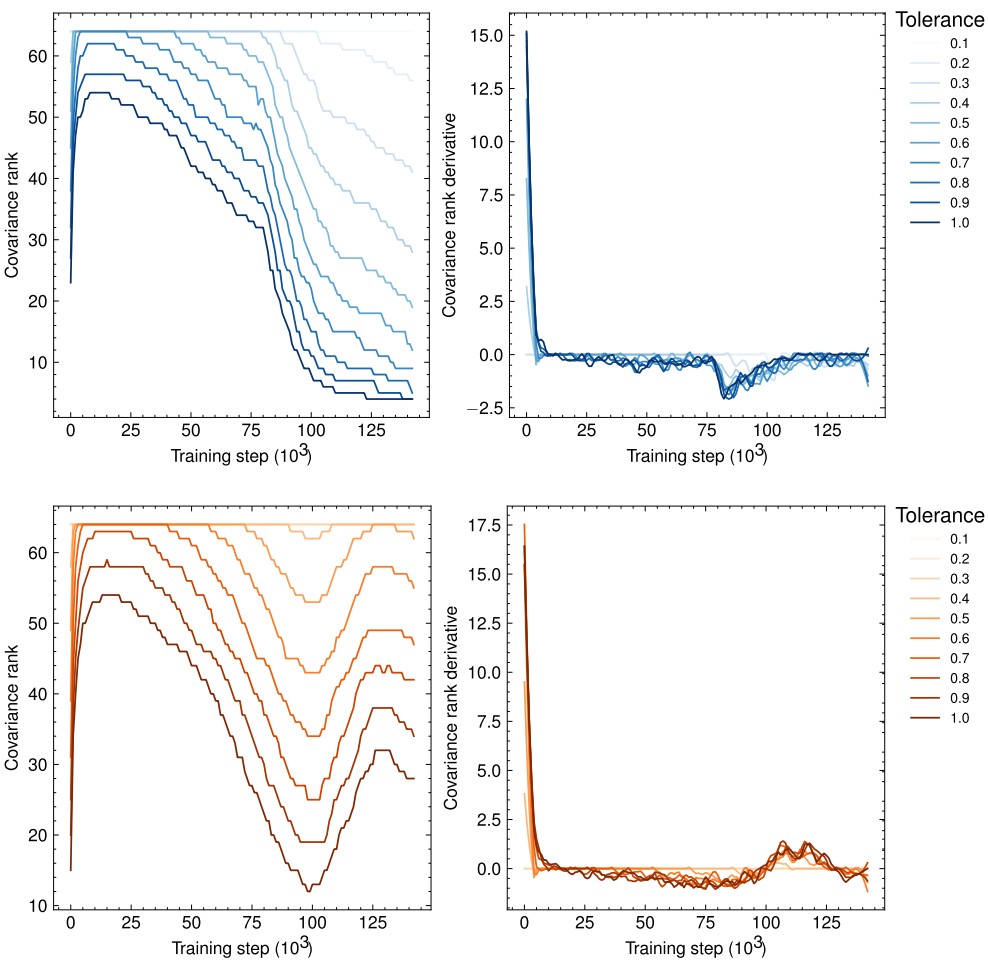

Figure 7: The covariance rank (left column) and its first derivative (right column) for the 70M model (top row) and for the 160M model (bottom row).

## 5 Conclusions

In this study we have analyzed both the temporal and spatial dimensions of training a large language model. As discussed above, our work is the first one dealing with distribution of network weights as a whole, by means of computational methods borrowed from statistical mechanics.

More specifically, this work shows that a bifurcation occurs in the dynamics of the weights during the training process. Such transitions are observed across various models of different sizes trained with distinct datasets. We have conducted a thorough and meticulous analysis of this aspect and concluded that this bifurcation marks a transition to a stationary state, indicating that further training is unlikely to significantly alter the weight values. Thus, training can be efficiently terminated upon reaching such a stationary state. Moreover, our study has offered a possible interpretation of the bifurcation phenomenon in terms of model perplexity.

Just as in the early days of thermodynamics, when empirical observations drove technological advancements, we advocate for the development of Large Language Models (LLMs) to be grounded in the observation of their internal dynamics. The identification of stationary states in the weight dynamics exemplifies this philosophy, marking a step toward a more observational and theoretically informed approach to LLM development.

Intriguingly, we finally note how recent works in the physics of complex networks point at diversity of information pathways as the main driver of sparsity in real networks Ghavasieh & De Domenico (2024): in this sense, we hypothesize that the poli-semanticity of natural language may act as the main driving force for network self-sparsification.

The presented results can have far-reaching implications as they demonstrate that keeping track of the collective behaviour of network weights could be a powerful indicator of training convergence, as opposed to the classical methods based on evaluation metrics which suffer from the confounding effects of non-linearity, hence giving raise to false claims about "emergent" properties. As a future work, we would like to further investigate the training loss dynamics, to check whether the sudden changes in quantitative microscopic parameters align with the development of *induction heads* in the model, as shown by (Bietti et al., 2023).

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
