# OpenReview forum: "The Garden of Forking paths: Observing Dynamic Parameters Distribution in Large Language Models"
_TMLR — Rejected by TMLR_

### Review · Reviewer_CugK · 2024-04-04

**Summary Of Contributions:**

The presented paper analyses the training dynamics of the Pythia checkpoints of varying model sizes and finds a two-stage training dynamics in the models readout layer.

**Audience:**

Yes

**Claims And Evidence:**

No

**Requested Changes:**

see weaknesses.

**Strengths And Weaknesses:**

#### **Strengths**

- Analysis of training dynamics for the publicly available checkpoints of Pythia models with varying sizes.
- Interesting finding that training parameters undergo two training stages: first one is diffusive, the second one is bimodal quasi-deterministic.
- Some experiments to verify these findings on a range of different sizes (14M, 31M, 70M, 160M) and datasets (deduped and non-deduped).

#### **Weaknesses**

- **[large]:** Both Figure 4 and Figure 6 show that for larger models (here 1B) this sudden drop is not present, suggesting that larger models have different training dynamics and challenging large portions of the presented work. Would be interesting to see similar experiments for the billion-level Pythia models (2.8B, 6.9B, 12B) as well as Figure 2 + Figure 3. I don't really see the point of computational cost as the checkpoints are available?
- **[medium]:** The proposal to conclude training once a stationary state is achieved seems to be already solved by simply monitoring validation set metrics and stopping based on validation convergence. Also, there's no analysis on how this proposal would work in practice and impact on actual metrics or experiments (including down-stream performance).

#### **Questions**

- Can you explain what exactly is visualized in Figure 2? I do understand that it is the readout (unembedding) layer of 4 models but how is the "density" computed?

---

### Review · Reviewer_JKsZ · 2024-04-08

**Summary Of Contributions:**

This paper empirically studies the dynamics of the parameters in the unembedding layer of small Pynthia transformer models throughout training. It shows that the distribution of parameters becomes bimodal at some timescale, which varies with the model size. Close to this bifurcation, a drop in the perplexity of the models on the LAMBADA dataset is observed. Moreover, the rank of the unembedding layer covariance decreases, suggesting that this layer maps the internal representation into a subspace of the complete vocabulary, thereby restricting the number of possible next tokens.

**Audience:**

No

**Claims And Evidence:**

No

**Requested Changes:**

In the present form, the paper has major technical flaws, limited evaluation, inadequate reproducibility, and requires some rewriting.

Beyond clarifying the concerns above, especially regarding the adopted definition of MSD and the experimental perplexity evaluation, I believe that to make the findings robust, it is necessary to consider:
- models of larger size (e.g., up to the B scale and with consistent datasets, for which pre-training checkpoints are all already available for the studied Pynthia models),
- different architectures (e.g., dynamical analysis on BLOOM, Workshop et al. (2022), and static analysis at the end of the training of the unembedding layers of other commonly used language models to verify whether the distribution is bimodal),
- extend the evaluation to different benchmarks.

Moreover, I don’t get why the paper only focuses on the unmbedding layers, which are also not always present in current models:
- On the one hand, the current presentation should adhere to the provided content and, therefore, mention everywhere it is appropriate that this work does not study the temporal evolution of transformer parameters during training but only focuses on a single layer, namely the unembedding layer.
- On the other hand, extending the analysis to other layers would be interesting.

**Strengths And Weaknesses:**

**Strengths**

- The paper’s objective, which focuses on understanding the inner workings of transformers and explaining their decision-making process, addresses a timely and impactful problem.
- Experimental methodologies, typical of the natural sciences, hold a big potential in closing the gap between practical applications and theoretical understanding.
- To the best of my knowledge, the identification of the bifurcation effect in the unembedding layer is novel.

**Weaknesses**

- The scope of the empirical analysis is limited. The paper considers the parameters of only a single layer for a range of small model sizes, restricted to a single architecture trained on two variants of the same dataset. Additionally, the evaluation of perplexity is conducted on a single benchmark dataset. All these facts constrain the generalizability of the findings.
- In the context of diffusion, the mean square displacement (MSD) is defined as the average deviation of a particle position after a time $t$. Instead, the paper seems to use a different definition, which involves a cumulative over time when considering the weight evolution, which is unclear how to interpret. Specifically, the paper’s MSD appears to reach zero (“its own small value”?) precisely at the end of training. Considering the standard MSD, this would imply that all weights return to their initial value – which is not clearly the case in practice.
- The reported perplexity values on the LAMBADA dataset are as low as $10^{-18}$. These values seem beyond the current best technology by 18 orders of magnitude (cf., e.g., Brown et al., 2020 for GPT3 with 175B parameters)! On a side note, in Fig. 4, the horizontal black line is drawn at perplexity **one** and not zero as reported.
- Despite a significant emphasis on mechanistic interpretability, the paper does not offer interpretations or deep discussion/analyses of the identified bifurcation effect.
- The presentation and writing quality of the paper could be better. The abstract is largely uninformative and lacks essential information on the paper’s research, methodology, and findings. Similar issues extend to the introduction. Some sections would benefit from more details, e.g., the training and evaluation datasets are not described. Figures are also poor, lacking necessary elements like axes labels or legends (e.g., in Fig. 2, values are missing in the y-axis, and similarly, the color scare is unknown) and sometimes are not references in the text (e.g., what is the goal of Fig. 1 panels B, C, D?). Captions are not informative of the figure’s content (e.g., in Fig. 7, the caption mentions the “covariance” of a model and does not explain what the tolerance is). Additionally, the wording is often too strong and, is just personal taste, but I find the writing style overly verbose and inadequate for a paper.
- Several claims lack verification or logical substantiation. For instance, the fact that models explore the landscape isotropically in a number of dimensions equals to the number of parameters (actually, isn’t the small value of the covariance of the unembedding weights suggesting something different?). Furthermore, in Sec. 4, the authors hypothesize that the bifurcation behavior will appear for larger models. However, they claim they cannot check, so they “reasonably believe their hypothesis is valid”. Furthermore, the paper claims (already in the abstract) to empirically show the reason behind the effectiveness of weight sparsification. Such a claim is again not scientifically demonstrated in this paper. First, the mentioned paper on quantization (Ma et al., 2024) quantized “every single parameter” and not just a small fraction of them, as in this paper. Second, to make such a claim even for only the embedding layers, I would have expected some empirical evidence, e.g., by binarizing the values of the embedding layer and evaluating the effect on the model output.
- The concept of mechanistic interpretability needs to be appropriately introduced and credited. In particular, it was not introduced in Conmy et al., 2023 (see references in the introduction of Conmy et al., 2023).
- Certain purely technical details, e.g., Sec. 3.2.1, could be more appropriately moved to the supplementary material.
- It is unclear why the proposed exit training strategy based on the bifurcation should be preferred to track the perplexity. Moreover, it seems that already for the larger models considered in this work – which are nevertheless orders of magnitude smaller compared to current language models – the effect is considerably delayed, raising doubts about its actual significant practical relevance.
- The parallelism with excitatory and inhibitory neurons in biology and the paper’s “potential biological relevance” is unclear.
- Overall, the paper would benefit from a more focused approach, prioritizing scientific rigor and empirical validation over speculative assertions of significant practical applications and “profound and far-reaching implications”.

Brown, T., Mann, B., Ryder, N., Subbiah, M., Kaplan, J. D., Dhariwal, P., ... & Amodei, D. (2020). Language models are few-shot learners. *Advances in neural information processing systems*, 33, 1877-1901.

*Typos*

Page 1: a ensemble (an ensemble)

Page 2: dynamic (dynamics)

Page 7: brownian (Brownian)

Reference Ma et al. (2024) is repeated twice in the References section.

---

### Review · Reviewer_U9tt · 2024-05-01

**Summary Of Contributions:**

The paper studies the time evolution of the statistic distribution of model parameters in Large Language Models (LLMs). The authors focus on the unembedding parameters which project the model's logits into the vocabulary space. The proposed analysis is based on Pythia, for which checkpoints of models of different size at multiple training stages are available. This allows the authors to monitor the evolution of the model's parameters as training progresses. The main finding of the paper is that the unembedding matrix dynamics witness a bifurcation effect, transitioning from a diffusive regime to a bimodal quasi-determinstic one. The transition point between the two regimes is empirically shown to depend on the model size. The authors then study the effects of this bifurcation point on model performance, measured via perplexity. When the bifurcation takes place, the models exhibit a sharp decrease in perplexity, suggesting that training can heuristically been stopped as soon as the transition occurs.

**Audience:**

Yes

**Broader Impact Concerns:**

The work does not present any negative ethical implications.

**Claims And Evidence:**

Yes

**Requested Changes:**

Generally, I think the analysis should be broadened and made more general. While the bifurcation effect identified by the authors is interesting, its causes, consequences and level of generality should be studied more thoroughly. In particular,

1. Does the same phenomenon happens in other models?
2. Do the authors have any insight on the causes of this phenomenon?
3. Is the bifurcation affecting only unembedding parameters or also other parts of the model?
4. Is it actually the case that the emergence of the bifurcation can be exploited to early-stop training?

I believe it would be beneficial for the paper to take these points into consideration.

**Strengths And Weaknesses:**

**Strenghts**
1. The paper is generally well written and easy to follow and the related literature is adequately covered.
2. The bifurcation effect identified by the authors is interesting and, to the best of the reviewer knowledge, novel.

**Weaknesses**
1. The bifurcation effect is showcased for only one model, Pythia. To quantify the degree of generality of this phenomenon, it would be interesting to understand if the same bifurcation effect takes place in other LLMs as well.
2. The analysis does not provide any motivations for the occurrence of the bifurcation effect. Why does this happen? Is it possible to predict at which phase of the training process the effect arises? How does it depend on the training dataset?
3. The analysis is restricted to unembedding parameters. Is there any visible effect taking place on other parts of the model when the bifurcation occurs?
4. It would be interesting to see how the model performance changes as a function of the training iterations on a wider set of downstream tasks. Is it the case that halting training right after the bifurcation occurs is enough to obtain a well-trained model? In other words, is the model still improving after the bifurcation?
5. Fig.1 A. illustrates a different model compared to the one described in Eq. 3. In the latter, attention and feedforward layers are calculated in parallel (following Pythia and GPT-NeoX), while in the figure the are computed sequentially (as in standard transformers).

---

### Decision · Action_Editor_z22P · 2024-07-16

**Recommendation:** Reject

**Comment:**

The paper addresses an interesting and timely problem, showing potential for a significant contribution to understanding transformer dynamics. However, to reach the standards expected by TMLR, the paper needs broader analysis across multiple models and layers, clearer explanations for observed phenomena, and more robust empirical validation. Improving the presentation and addressing reviewers' feedback will strengthen the paper considerably. I encourage the authors to expand their scope, and refine their analysis, to reach a more comprehensive and polished version.

**Audience:**

Simialrly, the concensus among reviewers is that, while the topic is of potential interest, the paper's slightly narrow focus, speculative practical applications, and presentation issues limit its appeal. The findings mat not be generalizable, and the suggestions for practical applications are found to be not sufficiently backed by evidence.

**Claims And Evidence:**

Reviewers all found that the claims made in the submission are not adequately supported by accurate, convincing, and clear evidence. The analysis is limited to one model, lacks explanations for the observed bifurcation effect, and fails to provide sufficient empirical validation for early stopping based on bifurcation points. Additionally, there are inconsistencies and errors in the definitions and interpretations used, and the authors did not address reviewers' concerns, further weakening the submission.

**Resubmission Of Major Revision:**

The authors may consider submitting a major revision at a later time.